# BLOCKVID: BLOCK DIFFUSION FOR HIGH-FIDELITY AND COHERENT MINUTE-LONG VIDEO GENERATION

## ABSTRACT

Generating minute-long videos is a critical step toward developing world models, providing a foundation for realistic extended scenes and advanced AI simulators. The emerging semi-autoregressive (block diffusion) paradigm integrates the strengths of diffusion and autoregressive models, enabling arbitrary-length video generation and improving inference efficiency through KV caching and parallel sampling. However, it still faces challenges such as error accumulation from KV caching over long sequences and the absence of suitable evaluation benchmarks. To overcome these limitations, we propose *BlockVid*, a novel block diffusion framework equipped with a semantic-aware sparse KV cache, an effective training strategy called Block Forcing, and dedicated noise scheduling to reduce error propagation and enhance temporal consistency. Additionally, we introduce *LV-Bench*, a fine-grained benchmark for minute-long videos, complete with new metrics designed to evaluate long-range coherence. Extensive experiments on VBench and LV-Bench demonstrate that our approach consistently outperforms existing methods in generating high-quality, coherent minute-long videos. In particular, it achieves a **22.2%** improvement on VDE Subject and a **19.4%** improvement on VDE Clarity in LV-Bench over the current state of the art.

## 1 INTRODUCTION

Long video generation is crucial for creating realistic and coherent narratives that unfold over extended durations, which is essential for applications such as filmmaking, digital storytelling, and virtual simulation (Yi et al., 2025; Wang et al., 2025; Huang et al., 2024a; Liu et al., 2025). Moreover, the ability to generate minute-long videos is a key step toward building world models, which act as foundational simulators for agentic AI, embodied AI, and gaming (Che et al.; Shi et al., 2025).

A key breakthrough empowering this is the semi-autoregressive (block-diffusion) decoding paradigm, as shown in Figure 1 (3) (Arriola et al., 2025), which merges the strengths of diffusion and autoregressive methods by generating video tokens in blocks—applying diffusion within each block while conditioning on previous ones, resulting in more coherent and stable

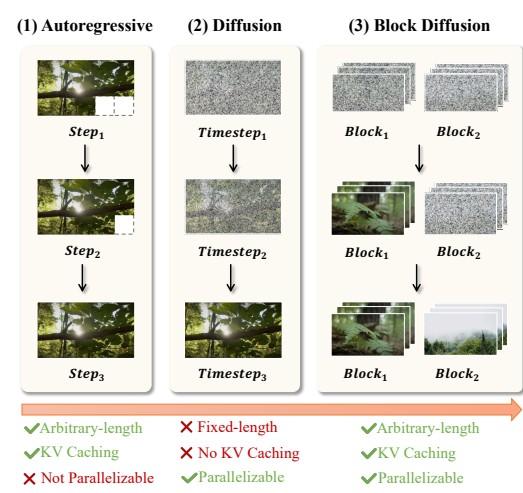

**(1) Autoregressive**    **(2) Diffusion**    **(3) Block Diffusion**

Figure 1: **Architecture comparison:** AR vs. Diffusion vs. Block Diffusion (Semi-AR).

video sequences (Huang et al., 2025; Teng et al., 2025). Notably, it addresses the key limitations of both diffusion and autoregressive (AR) models. Most current video diffusion models (Wan et al., 2025) rely on the Diffusion Transformer (DiT) (Peebles & Xie, 2023)—which uses bidirectional attention without KV caching. While this enables parallelized generation and controllability, decoding is inefficient and restricted to fixed lengths. In contrast, AR-based frameworks (Wang et al., 2024) support variable-length generation and KV Cache management, but their generation quality lags behind video diffusion, and decoding is not parallelizable. Importantly, block diffusion (Huang

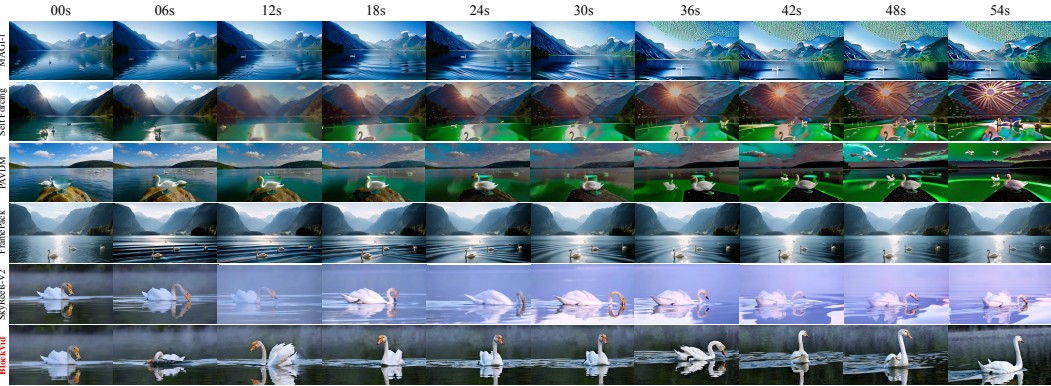

***Prompt summary:*** *A graceful white swan glides across a misty lake, dipping, splashing, and turning with serene elegance, its reflection mirroring every movement in the calm water.*

Figure 2: Comparison of visualization results between our method and different baselines in terms of accumulation error. Details can be found in Appendix A.7.

et al., 2025; Teng et al., 2025) interpolates between AR and diffusion by reintroducing LLM-style KV Cache management, enabling efficient, variable-length, and high-quality generation.

However, existing block diffusion methods face two fundamental challenges. First, AR models inevitably suffer from error accumulation, where small prediction mistakes gradually build up over time and will be directly stored in the KV cache (Kang et al., 2024). In long video generation, these accumulated errors typically manifest as quality degradation, color drift, subject and background inconsistency, and visual distortions (Lu et al., 2024). As the sequence extends, these errors compound, weakening long-range dependencies and limiting its effectiveness for generating coherent, minute-long videos (see Figure 2). Second, the domain is hindered by *the lack of fine-grained long video datasets and reliable evaluation metrics*. Currently, most open-source datasets consist of only short or fragmented chunks, with few minute-long datasets featuring fine-grained annotations. Meanwhile, existing benchmarks and metrics like VBench (Huang et al., 2024b) focus on diversity or object categories but fail to capture error accumulation and coherence over extended durations.

To this end, we propose *BlockVid*, a semi-autoregressive block diffusion model generating minute-long videos in a chunk-by-chunk manner, as shown in Figure 3. Three strategies are proposed to systematically address the accumulation error induced by the KV cache from both training and inference perspectives. 1) We first introduce a semantic sparse KV cache that selectively stores salient tokens from past chunks and retrieves the most semantically aligned context for the current prompt, thereby maintaining long-range consistency without propagating redundant errors. 2) Moreover, to bridge the training–inference gap, we introduce *Block Forcing*, which combines Velocity Forcing loss to regularize chunk-wise predictions with Self Forcing loss (Huang et al., 2025). This prevents models from drifting over long horizons, such as losing track of subjects or gradually altering scene content. 3) We further explore various noise scheduling and shuffling strategies for long video generation to enhance temporal consistency and reduce error accumulation over extended durations.

To address the lack of long-video datasets and benchmarks, we propose *LV-Bench*, a collection of 1,000 minute-long videos with fine-grained annotations for every 2–5 second chunk. To better evaluate long video generation quality, we further introduce Video Drift Error (VDE) metrics based on Weighted Mean Absolute Percentage Error (WMAPE) (Kim & Kim, 2016; De Myttenaere et al., 2016), integrated with original VBench metrics, providing a more comprehensive reflection of temporal consistency and long-range visual fidelity.

Comprehensive experiments are conducted on both LV-Bench and the traditional VBench to demonstrate the superiority of our method. *BlockVid* achieves a **22.2%** improvement on VDE Subject and a **19.4%** improvement on VDE Clarity in LV-Bench compared to the current state-of-the-art method.

## 2 RELATED WORK

**Long video generation.** Minute-long video generation can be grouped into three settings: single-shot video generation, multi-shot video generation, and movie-style video composition.

(1) Single-shot generation aims to produce a minute-long chunk within a consistent scene and semantic context, emphasizing long-range temporal coherence and visual stability. Approaches fall

into autoregressive (AR) and semi-autoregressive (semi-AR, i.e., block diffusion) families. AR methods, such as FAR (Gu et al., 2025) and Loong (Wang et al., 2024), formulate long video generation as next-frame (or next segment) prediction. Semi-AR methods generate videos chunk by chunk, while performing iterative diffusion-based (Peebles & Xie, 2023) denoising within each chunk. Their key design choice lies in the chunk-level causal conditioning: MAGI-1 (Teng et al., 2025), Skyreel-V2 (Chen et al., 2025), and Self Forcing (Huang et al., 2025) proceed strictly sequentially across chunks, whereas FramePack (Zhang & Agrawala, 2025) adopts a symmetric schedule that treats both ends as guidance and fills the middle autoregressively. In practice, semi-AR methods typically rely on *careful KV cache usage* for efficiency and stability over long horizons.

(2) Multi-shot generation typically focuses on handling camera motions and transitions across scenes or semantics. Recent systems, such as LCT (Guo et al., 2025), RIFLEx (Zhao et al., 2025), and MoC (Cai et al., 2025), often organize text–video units with interleaved layouts and positional extrapolation to accommodate multiple shots.

(3) Movie-style generation aims to create cinematic content by stitching together multiple chunks, often with different scenes and styles, while maintaining a coherent global narrative or theme. Methods such as VideoTTT (Dalal et al., 2025), MovieDreamer (Zhao et al., 2024), MovieBench (Wu et al., 2025), and Captain Cinema (Xiao et al., 2025) resemble film editing, combining diverse shots into a single coherent video guided by chunk-level text descriptions.

**Block diffusion** (also called semi-autoregressive or chunk-by-chunk diffusion) decodes long sequences in blocks: within each block the model performs iterative diffusion denoising, while across blocks it conditions causally on previously generated content via KV caches. This paradigm has been explored in both text and video. In language modeling, BD3-LM (Arriola et al., 2025) and SSD-LM (Han et al., 2022) demonstrate that blockwise diffusion can combine bidirectional refinement within a block with efficient, variable-length decoding through cached context across blocks. In video generation, related formulations adopt chunk-wise diffusion with causal conditioning to interpolate between pure diffusion (e.g., DiT-style bidirectional attention without KV caching) and autoregression (variable-length decoding with KV caching but weaker visual fidelity and limited parallelism). Representative systems include MAGI-1 (Teng et al., 2025), Self Forcing (Huang et al., 2025), CausVid (Yin et al., 2025), ViD-GPT (Gao et al., 2024), and SkyReels-V2 (Chen et al., 2025), which condition each new chunk on past chunks to extend temporal horizons while retaining diffusion's denoising quality within a chunk. Despite progress, block diffusion methods remain constrained by KV cache–induced errors, limited scalability, and the lack of long video datasets and coherence-aware metrics. We address these gaps with (1) BlockVid, a framework featuring semantic sparse KV cache, Block Forcing, and tailored noise scheduling to enhance long-range coherence, and (2) LV-Bench, a benchmark of 1,000 minute-long videos with metrics for evaluating temporal consistency.

## 3 METHOD

### 3.1 OVERVIEW: BLOCK DIFFUSION ARCHITECTURE

BlockVid introduces a semi-AR block diffusion architecture. During training, we are given a single-shot long video $V = \{V_1, V_2, V_3, \ldots, V_n\}$, where each video chunk $V_i \in \mathbb{R}^{(1+T) \times H \times W \times 3}$, with $T$ frames, height $H$, width $W$, and 3 RGB channels. We also have the corresponding chunk level prompts $\mathcal{Y} = \{y_i\}_{i=1}^n$, with $y_i$ conditioning $V_i$. Specifically, the first frame serves as the image guidance. The 3D causal VAE compresses its spatio-temporal dimensions to $[(1+T/4), H/8, W/8]$ while expanding the number of channels to 16, resulting in the latent representation $Z \in \mathbb{R}^{(1+T/4) \times H/8 \times W/8 \times 16}$. The first frame is compressed only spatially to better handle the image guidance.

During post-training, we introduce Block Forcing, a training strategy that stabilizes long video generation by jointly integrating Velocity Forcing and Self Forcing objectives. Velocity Forcing aligns predicted dynamics with semantic history to prevent drift, while Self Forcing closes the training–inference gap by exposing the model to its own roll-outs and enforcing sequence-level realism.

As shown in Figure 3, in the latent space, the representation $Z$ is first processed by the block diffusion denoiser to produce the denoised latent $\tilde{Z}$. During this procedure, the semantic sparse KV cache is dynamically constructed and preserved as a compact memory of salient keys and values,

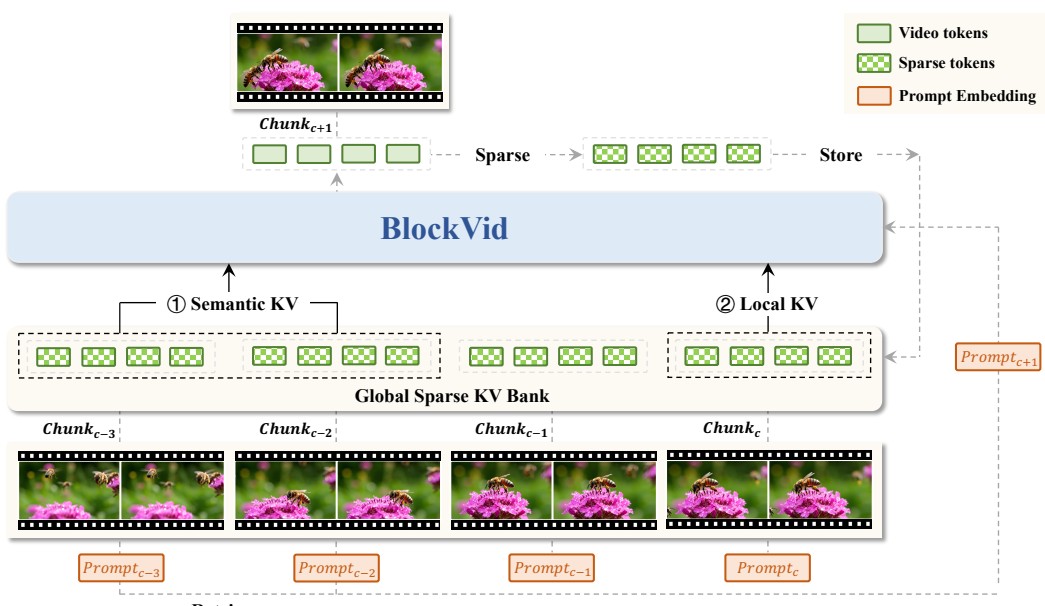

Figure 3: **Overview of the BlockVid semi-AR framework.** The generation of chunk $c+1$ is conditioned on both a local KV cache and a globally retrieved context. The global context is dynamically assembled by retrieving top-$l$ semantically similar KV chunks via prompt embedding similarity. Upon generation, the bank is updated with the new chunk's most salient KV tokens.

serving as semantic guidance for subsequent chunk generation. Subsequently, the denoised latent $\tilde{Z}$ is projected back into the video space $\tilde{X}$.

Besides, we design a noise scheduling strategy that operates both during training and inference to stabilize long video generation. During training, progressive noise scheduling gradually increases noise levels across chunks. While during inference, noise shuffling introduces local randomness at chunk boundaries to smooth transitions and maintain coherence.

### 3.2 BLOCK FORCING

**Self Forcing.** A major challenge in long video generation is the *training-inference gap*: during training the model is conditioned on ground-truth frames (teacher forcing), but at inference it must rely on its own imperfect outputs, leading to exposure bias and error accumulation. To address this, we adopt the *Self Forcing* loss (Huang et al., 2025), where the model generates a full video sequence $\tilde{x}_{1:T}$ semi-autoregressively and is then penalized at the *video level* by matching its distribution $p_\theta$ to the real data distribution $p_{\text{data}}$. Concretely, a discriminator $D$ evaluates entire videos, and the generator $G$ is trained to minimize

$$\mathcal{L}_{\text{SF}} = \min_G \max_D \ \mathbb{E}_{x \sim p_{\text{data}}}[\log D(x)] \ + \ \mathbb{E}_{\tilde{x} \sim p_\theta}[\log(1 - D(\tilde{x}))], \tag{1}$$

where $\tilde{x} \sim p_\theta$ is obtained by the predictions of $G$. This formulation exposes the model to its own errors during training and enforces sequence-level realism, thereby reducing exposure bias and improving temporal consistency in long videos.

**Velocity Forcing.** When generating very long videos, the model trained with Self Forcing alone can lose track of the subject or scene, leading to drift (e.g., the character slowly changing identity or the background gradually melting). To address this problem, we introduce a *Velocity Forcing* loss, decomposes the learning objective into two complementary parts. From a fidelity perspective, Velocity Forcing ensures the reconstruction accuracy of the current video chunk. This is determined by the ground-truth starting frame $x_{\text{start}}$ of the current chunk and the corresponding Gaussian noise $\epsilon$, which together define the standard diffusion training signal. From a semantic perspective, Velocity Forcing enforces semantic alignment between the current video chunk and its most relevant historical context. Specifically, the top-$l$ past chunks are resampled to match the temporal length of

the current chunk and averaged into a semantic reference $x_{\text{cond}}$, which serves as high-level guidance to maintain long-term coherence.

In the stochastic interpolant formulation of flow matching (Albergo & Vanden-Eijnden, 2022), the model predicts a velocity field $v_{\text{pred}}$ that represents the temporal derivative of the interpolated state $x_t$ between noise and data.

$$v_{\text{pred}} = v_t(x_t) = \frac{d}{dt}x_t = f_\theta(x_t, t). \tag{2}$$

Formally, the Velocity Forcing loss penalizes the deviation of the predicted velocity $v_{\text{pred}}$ from both the noise term $\epsilon$ and the semantic reference $x_{\text{cond}}$, weighted by a coefficient $\gamma \in [0, 1]$:

$$\mathcal{L}_{\text{VF}} = \mathbb{E}\Big[\|v_{\text{pred}} - (\epsilon - \gamma \cdot x_{\text{cond}})\|^2\Big]. \tag{3}$$

This formulation ensures that the model learns not only to denoise the current chunk correctly but also to remain semantically anchored to the relevant history, thereby reducing temporal drift and improving the stability of long video generation. The final training loss is $\mathcal{L} = \mathcal{L}_{\text{SF}} + \mathcal{L}_{\text{VF}}$.

### 3.3 Semantic Sparse KV Cache

To efficiently preserve long-range temporal dependencies, we introduce a *Semantic Sparse KV Cache* that stores and reuses key-value pairs across video chunks. Inspired by ZipVL (He et al., 2025), we first dynamically identify salient tokens with a probing mechanism and store the most informative KV tokens as the KV cache.

Formally, given the current chunk $c$ and its queries $Q$, keys $K$, and values $V$, we compute the attention score matrix

$$A = \text{Softmax}\Big(\frac{QK^\top}{\sqrt{d}} + \text{MASK}\Big), \tag{4}$$

where the MASK denotes a chunk-level causal attention mask.

Then aggregate scores across heads and probe queries to form an importance vector $\mathbf{m}$. Then the important tokens are selected using the top-$k$ indexing method, with $M$ be the minimal number of tokens that cover a fraction $\tau$ of the total importance score:

$$\mathcal{I}_{\text{keep}} = \text{topk\_index}(m, M). \tag{5}$$

This produces a sparse cache $(K_{\text{sparse}}, V_{\text{sparse}})$ containing only the most relevant context tokens.

During long video generation, the sparse KV caches from past chunks are stored in a global KV bank and retrieved based on the semantic similarity of prompt embeddings:

$$\text{sim}_i = \cos\big(E_c, E_i\big), \quad i \in \{1, \ldots, c-1\}, \tag{6}$$

where $E_c$ is the embedding of the current prompt and $E_i$ are embeddings of past prompts. The top-$l$ most similar entries are then selected. Finally, we concatenate the top-$l$ semantic KV caches with the two most recent caches to form the final KV cache:

$$(K^*, V^*) = \text{CONCATKV}\Big(\{(K_j, V_j)\}_{j \in \text{seq\_ctx}}, \{(K_i, V_i)\}_{i \in \text{top-}l}\Big), \tag{7}$$

where $\text{seq\_ctx} = \{c-3, c-2\}$ (if available). The detailed algorithm is provided in Appendix A.2.

Finally, the aggregated KV cache $(K^*, V^*)$ serves as conditional context, combined with the current prompt $y_t$ to guide the generation of the target chunk:

$$V_t \sim p_\theta(\cdot \mid K^*, V^*, y_t). \tag{8}$$

### 3.4 Noise Scheduling

**Progressive noise scheduling.** The core idea here is to assign each chunk a different noise level, progressively increasing noise levels rather than using a fix one. Specifically, if we split a video $V$ into $n$ chunks, each chunk is assigned a noise level $\epsilon_c$ increasing with $c$, where $c = 1, \ldots, n$.

Table 1: Overview of the datasets used for constructing LV-Bench.

| Dataset | Video Number | Object Classes |
|---|---|---|
| DanceTrack | 66 | Humans (66, 100%) |
| GOT-10k | 272 | Humans (177, 65%)  Animals (54, 20%)  Environment (41, 15%) |
| HD-VILA-100M | 117 | Humans (47, 40%)  Animals (35, 30%)  Environment (35, 30%) |
| ShareGPT4V | 545 | Humans (381, 70%)  Animals (82, 15%)  Environment (82, 15%) |
| **LV-Bench** | **1000** | **Humans (671, 67%)  Animals (171, 17%)  Environment (158, 16%)** |

We adopt a *cosine schedule*, which provides smooth acceleration and deceleration:

$$\epsilon_c = \epsilon_{\min} + \tfrac{1}{2}\big(\epsilon_{\max} - \epsilon_{\min}\big)\Big(1 - \cos\big(\pi \tfrac{c}{n-1}\big)\Big), \qquad c = 1, 2, \ldots, n. \tag{8}$$

In this setting, the first chunk has $\epsilon_0 = \epsilon_{\min}$ (nonzero initial noise), and the last chunk has $\epsilon_{n-1} = \epsilon_{\max}$ (maximal noise). For more noise schedules, please refer to Appendix A.5.

This progressive schedule helps mitigate error accumulation: the clean early chunks establish the scene, and noisier later chunks are guided by them. In fact, progressively increasing noise encourages later chunks to follow the patterns of the earlier and more certain frames, facilitating smoother temporal transition (Xie et al., 2025a). In other words, earlier chunks act as anchors, and the rising noise in later chunks ensures new frames remain consistent with the established content.

**Noise shuffling.** Inspired by FreeNoise (Qiu et al., 2023), we adapt local noise shuffling to the chunk-by-chunk setting. During inference, each chunk $c$ inherits per-frame base noises $\{\epsilon_t^{(c)}\}_{t=1}^{T}$ from a fixed random seed, where $t \in \{1, \ldots, T\}$ indexes frames within a chunk and $T$ is the number of frames per chunk. To smooth the transition across chunk boundaries, we apply a *shuffle unit* of size $s$ to the prefix and suffix regions. Specifically, the last $s$ frames of chunk $c$ and the first $s$ frames of chunk $c+1$ are shuffled independently within their local window:

$$\tilde{\epsilon}_{T-s+1:T}^{(c)} = \text{SHUFFLE}\big(\epsilon_{T-s+1:T}^{(c)}\big), \qquad \tilde{\epsilon}_{1:s}^{(c+1)} = \text{SHUFFLE}\big(\epsilon_{1:s}^{(c+1)}\big). \tag{9}$$

This local permutation preserves the global order of chunks while introducing shared stochasticity at the boundaries, which encourages the model to fuse adjacent chunks more smoothly. In contrast to re-sampling entirely new noise for each chunk, this strategy maintains long-range coherence while mitigating abrupt transitions at chunk boundaries.

## 4  LV-BENCH

**Dataset.** To tackle the challenge of minute-long video generation, we curate a dataset of 1000 videos from diverse open-source sources and annotate them in detail. As shown in Table 1, we collect high-quality video chunks with lengths of at least 50 seconds from DanceTrack (Sun et al., 2022), GOT-10k (Huang et al., 2019), HD-VILA-100M (Xue et al., 2022), and ShareGPT4V (Chen et al., 2024a). To obtain high-quality annotations, we employ GPT-4o as a data engine to generate fine-grained captions for every 2–3 seconds in each video. The detailed prompt can be found in Appendix A.3. Human-in-the-loop validation consists of manual visual checks at every stage of data production, including data sourcing, chunk splitting, and captioning, to ensure high-quality annotations. In the data sourcing stage, human annotators select high-quality videos and determine whether each raw video is suitable for inclusion. In the chunk splitting stage, human annotators examine samples to verify that each chunk is free from errors such as incorrect transitions. In the captioning stage, human annotators review the generated descriptions to ensure semantic accuracy and coherence. At each stage, at least two human annotators participate to provide inter-rater reliability. We then randomly divided LV-Bench into an 8:2 split for training and evaluation.

**Metrics.** Drift penalties have been widely adopted to address information dilution (Li et al., 2025) and degradation (Lu et al., 2024) in long video generation. For example, IP-FVR (Han et al., 2025) focuses on preserving identity consistency, while MoCA (Xie et al., 2025b) employs an identity perceptual loss to penalize frame-to-frame identity drift. Inspired by the Mean Absolute Percentage Error (MAPE) and Weighted Mean Absolute Percentage Error (WMAPE) (Kim & Kim, 2016; De Myttenaere et al., 2016), we propose a new metric called Video Drift Error (VDE) to measure changes in video quality. We further design 5 long video generation metrics based on VDE. The

Table 2: **Comparison of different methods on LV-Bench.** We report LV-Bench results on five VDE metrics and five complementary metrics from VBench (Huang et al., 2024b). Our method achieves superior performance on the majority of these metrics.

| Method | VDE Subject ↓ | VDE Background ↓ | VDE Motion ↓ | VDE Aesthetic ↓ | VDE Clarity ↓ |
|---|---|---|---|---|---|
| MAGI-1 | 0.3090 | 0.5000 | 0.0243 | 3.8286 | 2.7225 |
| Self Forcing | 0.3716 | 1.6108 | 0.1549 | 3.4683 | 3.0798 |
| PAVDM | 1.8292 | 0.9323 | 0.0461 | 2.8957 | 1.9503 |
| FramePack | 4.3984 | 5.9421 | 0.0387 | 1.4751 | 4.2513 |
| SkyReels-V2-DF-1.3B | 0.1085 | 0.3179 | 0.0195 | 1.2083 | 0.9365 |
| **BlockVid-1.3B (Ours)** | **0.0844** | **0.2945** | **0.0119** | **0.9618** | **0.7551** |

| Method | Subject Consistency ↑ | Background Consistency ↑ | Motion Smoothness ↑ | Aesthetic Quality ↑ | Image Quality ↑ |
|---|---|---|---|---|---|
| MAGI-1 | 0.8992 | 0.9078 | 0.9947 | **0.6508** | 0.6662 |
| Self Forcing | 0.8481 | 0.8203 | 0.9947 | 0.6283 | 0.6805 |
| PAVDM | 0.8640 | 0.8924 | 0.9926 | 0.5267 | 0.6567 |
| FramePack | 0.9001 | 0.8791 | 0.9949 | 0.6043 | **0.6972** |
| SkyReels-V2-DF-1.3B | 0.9418 | 0.9579 | 0.9931 | 0.6035 | 0.6835 |
| **BlockVid-1.3B (Ours)** | **0.9597** | **0.9588** | **0.9956** | 0.6047 | 0.6852 |

core idea involves dividing a long video into multiple smaller segments, each evaluated according to specific quality metrics (such as clarity, motion smoothness, etc). Specifically, (1) *VDE Clarity* measures temporal drift in image sharpness, where creeping blur increases the score, while a low value indicates stable clarity over time. (2) *VDE Motion* measures drift in motion smoothness, where a low score indicates consistent dynamics without jitter or freezing. (3) *VDE Aesthetic* measures drift in visual appeal, where a low score indicates sustained and coherent aesthetics over time. (4) *VDE Background* measures background stability, where a low score indicates a consistent setting without drift or flicker over time. (5) *VDE Subject* tracks identity drift, where a low score indicates the subject remains consistently recognizable over time. Following previous works (Guo et al., 2025; Cai et al., 2025), we also include five complementary metrics from VBench (Huang et al., 2024b). The details are included in Appendix A.4.

## 5 EXPERIMENT

### 5.1 IMPLEMENTATION DETAILS

**LV-1.1M dataset.** To improve post-training data for semi-AR models, we introduce LV-1.1M, a private curated dataset of 1.1M long-take videos with fine-grained annotations. Each video is segmented into chunks, captioned with GPT-4o, and aligned into coherent storylines, providing reliable supervision for long video generation. For more details see Appendix A.6.

**Multi-stage post-training.** We adopt a two-stage post-training strategy. In *Stage 1*, we post-train BlockVid on LV-1.1M to enhance its ability to handle long-take videos with coherent semantics, large-scale motions, and diverse content. This stage focuses on improving temporal reasoning and narrative consistency under high-quality but heterogeneous video data. In *Stage 2*, we further post-train the model on the training split of LV-Bench, a dataset containing longer videos ($\geq$50s) compared to Stage 1, in order to enhance the model's extrapolation capability and align with the evaluation protocols. The detailed training setup can be found in Appendix A.9.

### 5.2 MAIN RESULTS

**Results on LV-Bench.** We first compare our method with several open-source long video generation baselines on LV-Bench, including MAGI-1 (Teng et al., 2025), Self Forcing (Huang et al., 2025), PAVDM (Xie et al., 2025a), FramePack (Zhang & Agrawala, 2025), and SkyReels-V2-DF-1.3B (Chen et al., 2025). As shown in Table 2, our BlockVid-1.3B consistently outperforms these methods across most VDE metrics and complementary metrics from VBench. In partic-

Table 3: **Comparison of different methods on VBench** (Huang et al., 2024b). We report VBench metrics of different methods following the single-shot long video generation setting (Guo et al., 2025; Cai et al., 2025). Our method achieves superior performance in the majority of these metrics.

| Method | Subject Consistency ↑ | Background Consistency ↑ | Motion Smoothness ↑ | Dynamic Degree ↑ | Aesthetic Quality ↑ | Image Quality ↑ |
|---|---|---|---|---|---|---|
| MAGI-1 | 0.8320 | 0.8931 | 0.9740 | 0.5537 | 0.5010 | 0.6120 |
| Self Forcing | 0.8211 | 0.9050 | 0.9799 | 0.6015 | 0.5130 | 0.6218 |
| PAVDM | 0.8415 | 0.9273 | 0.9769 | 0.6537 | 0.4970 | 0.6280 |
| FramePack | 0.9019 | 0.9450 | 0.9805 | 0.5715 | 0.5044 | 0.6381 |
| SkyReels-V2-DF-1.3B | 0.9391 | 0.9580 | 0.9838 | 0.6529 | 0.5320 | 0.6315 |
| LCT (MMDiT-3B) | 0.9380 | 0.9623 | 0.9816 | 0.6875 | 0.5200 | 0.6345 |
| MoC | 0.9398 | **0.9670** | 0.9851 | 0.7500 | 0.5547 | 0.6396 |
| **BlockVid-1.3B (Ours)** | **0.9410** | 0.9650 | **0.9870** | **0.7720** | **0.5839** | **0.6527** |

ular, BlockVid achieves the lowest error scores on all five VDE metrics, reducing subject drift, background inconsistency, motion degradation, and perceptual losses compared to strong baselines such as SkyReels-V2-DF-1.3B. On complementary VBench metrics, BlockVid also delivers the highest subject consistency (0.9597) and background consistency (0.9588), as well as superior motion smoothness (0.9956). Although SkyReels-V2-DF-1.3B attains slightly better aesthetic quality (0.6035 vs. 0.6047, lower is better) and FramePack yields marginally higher image quality (0.6972), BlockVid maintains competitive performance on these aspects while achieving state-of-the-art results overall. These results demonstrate that our method not only improves long-term coherence but also balances fidelity and aesthetics in long video generation.

**Results on VBench.** We further compare our method with state-of-the-art baselines on VBench (Huang et al., 2024b) under the single-shot long video generation setting (Guo et al., 2025; Cai et al., 2025). As shown in Table 3, BlockVid-1.3B achieves superior performance across the majority of metrics, surpassing both open-source and large-scale proprietary baselines. Specifically, BlockVid obtains the highest scores in subject consistency (0.9410), motion smoothness (0.9870), dynamic degree (0.7720), aesthetic quality (0.5839), and image quality (0.6527), demonstrating its ability to generate temporally coherent, visually appealing, and semantically dynamic long videos. While MoC slightly outperforms BlockVid in background consistency (0.9670 vs. 0.9650), our model delivers the most balanced overall performance across all six dimensions. These results highlight the effectiveness of BlockVid in achieving both temporal stability and perceptual quality in long video generation.

## 5.3 ABLATION STUDY

We further conduct ablation studies from four perspectives: *noise scheduling*, *KV cache settings*, *Block Forcing*, and *post-training datasets*, as detailed below.

**Noise scheduling.** As shown in Table 4, during post-training, the *cosine noise schedule* achieves the best overall performance compared to naive or alternative scheduling strategies. During inference, noise shuffle with a window size of $s = 4$ further enhances temporal smoothness across chunk boundaries, leading to the most stable and coherent long video generation.

Table 4: Ablation on noise schedule.

| Method | VDE Subject ↓ | VDE Background ↓ | VDE Motion ↓ | VDE Aesthetic ↓ | VDE Clarity ↓ |
|---|---|---|---|---|---|
| Naive | 0.0936 | **0.2894** | 0.2311 | 0.9643 | 0.7791 |
| Linear | 0.0935 | 0.3015 | 0.0167 | **0.8910** | 0.7610 |
| Cosine | **0.0844** | 0.2945 | **0.0119** | 0.9618 | **0.7551** |
| Sigmoid | 0.0961 | 0.4027 | 0.0276 | 0.9723 | 0.8247 |
| No Shuffle | 0.0902 | 0.3007 | 0.0281 | 0.9635 | 0.7580 |
| s=2 | 0.0853 | 0.2995 | 0.0138 | 0.9730 | **0.7492** |
| s=4 | **0.0844** | **0.2945** | **0.0119** | **0.9618** | 0.7551 |

**KV cache.** We further explore rolling KV (Huang et al., 2025), dynamic sparse KV (He et al., 2025), and our semantic sparse KV under different attention thresholds $\tau$. As shown in Table 5, our semantic sparse KV cache with $\tau = 0.98$ achieves the best overall performance, consistently reducing subject, background, motion, aesthetic, and clarity errors compared to other baselines.

Table 5: Ablation on KV cache settings.

| Method | VDE Subject ↓ | VDE Background ↓ | VDE Motion ↓ | VDE Aesthetic ↓ | VDE Clarity ↓ |
|---|---|---|---|---|---|
| Rolling KV | 0.0961 | 0.3519 | 0.0547 | 0.9815 | 0.7913 |
| Dynamic Sparse KV ($\tau = 0.97$) | 0.0927 | 0.3074 | 0.0253 | 0.9781 | 0.7730 |
| Dynamic Sparse KV ($\tau = 0.98$) | 0.0910 | 0.3040 | 0.0239 | 0.9716 | 0.7652 |
| **Semantic Sparse KV** ($\tau = 0.97$) | 0.0869 | 0.2988 | 0.0153 | 0.9684 | 0.7570 |
| **Semantic Sparse KV** ($\tau = 0.98$) | **0.0844** | **0.2945** | **0.0119** | **0.9618** | **0.7551** |

**Block Forcing.** Table 6 shows that combining Self Forcing (Huang et al., 2025) and Velocity Forcing yields the best results, with our full Block Forcing achieving the lowest errors across all VDE metrics.

Table 6: Ablation on Block Forcing.

| Method | VDE Subject ↓ | VDE Background ↓ | VDE Motion ↓ | VDE Aesthetic ↓ | VDE Clarity ↓ |
|---|---|---|---|---|---|
| Naive | 0.0910 | 0.3317 | 0.0259 | 0.9810 | 0.7835 |
| Self Forcing | 0.0885 | 0.3155 | 0.0169 | 0.9658 | 0.7630 |
| Velocity Forcing | 0.0861 | 0.3015 | 0.0137 | 0.9673 | 0.7618 |
| **Ours** | **0.0844** | **0.2945** | **0.0119** | **0.9618** | **0.7551** |

**Post-training datasets.** As shown in Table 7, Stage 2 training on LV-Bench provides significantly greater improvements than Stage 1 training on LV-1.1M, as long videos ($\geq$50s) offer crucial extrapolation benefits for minute-long generation. Furthermore, our multi-stage post-training proves essential for achieving the best overall performance.

Table 7: Ablation on post-training datasets.

| Method | VDE Subject ↓ | VDE Background ↓ | VDE Motion ↓ | VDE Aesthetic ↓ | VDE Clarity ↓ |
|---|---|---|---|---|---|
| Stage 1 only | 0.8891 | 1.1573 | 0.0491 | 1.3742 | 1.2463 |
| Stage 2 only | 0.1752 | 0.4722 | 0.0153 | 0.9946 | 0.8452 |
| **Stage 1 + 2** | **0.0844** | **0.2945** | **0.0119** | **0.9618** | **0.7551** |

## 6 CONCLUSION

In this work, we have introduced *BlockVid*, an effective block diffusion framework for minute-long video generation. Our design integrates three key innovations: a *semantic sparse KV cache* that selectively retrieves salient context to mitigate error accumulation, a *Block Forcing* strategy that combines Velocity Forcing and Self Forcing to reduce temporal drift and close the training–inference gap, and a *noise scheduling* scheme that stabilizes long-horizon generation. Together, these components enable BlockVid to significantly improve long-range temporal coherence while maintaining high visual fidelity. To address the absence of suitable evaluation resources, we have further proposed *LV-Bench*, a fine-grained benchmark of 1,000 minute-long videos with detailed chunk-level annotations. Alongside, we have introduced Video Drift Error (VDE) metrics, which directly quantify coherence degradation over time. Our extensive experiments on LV-Bench and VBench have demonstrated that BlockVid achieves state-of-the-art performance, outperforming prior open-source and proprietary baselines across both coherence-aware and perceptual quality metrics.

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

## A APPENDIX

### A.1 LLM USE DECLARATION

Large Language Models (ChatGPT) were used exclusively to improve the clarity and fluency of English writing. They were not involved in research ideation, experimental design, data analysis, or interpretation. The authors take full responsibility for all content.

### A.2 ALGORITHM: SEMANTIC SPARSE KV CACHE

---

**Algorithm 1** Semantic Sparse KV Cache

---

1: **Input:** chunks $\{X_i\}_{i=1}^N$, prompts $\{\mathcal{Y}_i\}_{i=1}^N$, target $t=N$, threshold $\tau$, top-$K$, drop $p_{drop}$
2: **Output:** final KV cache $(\mathsf{K}^*, \mathsf{V}^*)$ for $X_t$
3: KV_BANK $\leftarrow \varnothing$                                       $\triangleright$ dict: $i \mapsto (\mathsf{K}_{\text{sparse}}^{(i)}, \mathsf{V}_{\text{sparse}}^{(i)})$
   **// Stage A: Build and store sparse KV for all prior chunks**
4: **for** $c \in \{1, \ldots, N-1\}$ **do**
5:    **if** $c \notin$ KV_BANK **then**
6:       $(\mathsf{K}_{\text{sparse}}^{(c)}, \mathsf{V}_{\text{sparse}}^{(c)}) \leftarrow$ BUILDSPARSEKV$(X_c, \mathcal{Y}_c, \tau)$
7:       KV_BANK$[c] \leftarrow (\mathsf{K}_{\text{sparse}}^{(c)}, \mathsf{V}_{\text{sparse}}^{(c)})$
8:    **end if**
9: **end for**
   **// Stage B: Retrieve Top-$K$ semantic from bank (no recompute)**
10: seq_ctx $\leftarrow \{N-3, N-2\}$ (if available)
11: $E_t \leftarrow$ MEANEMBED$(\mathcal{Y}_t)$
12: $\mathcal{S} \leftarrow \{1, \ldots, N-1\} \setminus$ seq_ctx
13: **for** $i \in \mathcal{S}$ **do**
14:    $E_i \leftarrow$ T5-EMBED$(\mathcal{Y}_i)$; $sim_i \leftarrow \cos(E_t, E_i)$
15: **end for**
16: Top-l-Idx $\leftarrow$ argsort$(\{sim_i\}_{i\in\mathcal{S}})[-l :]$
17: $(\mathsf{K}_{\text{seq}}, \mathsf{V}_{\text{seq}}) \leftarrow$ CONCATKV$(\{$KV_BANK$[j] : j \in$ seq_ctx$\})$
18: $(\mathsf{K}_{\text{sem}}, \mathsf{V}_{\text{sem}}) \leftarrow$ CONCATKV$(\{$KV_BANK$[i] : i \in$ TopKIdx$\})$
    **// Stage C: Final merge (seq_ctx + Top-l semantic) & token drop**
19: $(\mathsf{K}^*, \mathsf{V}^*) \leftarrow$ CONCATKV$((\mathsf{K}_{\text{seq}}, \mathsf{V}_{\text{seq}}), (\mathsf{K}_{\text{sem}}, \mathsf{V}_{\text{sem}}))$
20: **return** $(\mathsf{K}^*, \mathsf{V}^*)$

---

**Algorithm 2** BuildSparseKV: Dynamic Sparse KV Cache

---

1: **function** BUILDSPARSEKV$(X, \mathcal{Y}, \tau)$
2:    $H \leftarrow$ ENCODE$(X, \mathcal{Y})$                                       $\triangleright$ model input states
3:    $Q, K, V \leftarrow$ PROJECT$(H)$;    $(Q, K) \leftarrow$ ROPE$(Q, K)$
4:    $q\_len \leftarrow$ length$(Q)$
5:    **if** $q\_len > 1$ **then**                                       $\triangleright$ prefill stage (identify salient keys)
6:       $\mathcal{I}_{\text{probe}} \leftarrow$ CONCAT(Recent(64), Random(64, range=$[0, q\_len-64)$))
7:       $Q_{\text{probe}} \leftarrow Q[:, \mathcal{I}_{\text{probe}}, :]$
8:       $A \leftarrow$ SOFTMAX$\left(\frac{Q_{\text{probe}} K^\top}{\sqrt{d}} +$ CAUSALMASK$(\mathcal{I}_{\text{probe}}, q\_len)\right)$
9:       $s \leftarrow \sum_{\text{heads,probe}} A$                                       $\triangleright$ aggregate over heads and probe queries
10:      $m \leftarrow$ CUMMEAN$(s)$                                       $\triangleright$ cumulative mean (older tokens discounted)
11:      $M \leftarrow$ COVERCOUNT$(m, \tau)$                                       $\triangleright$ smallest $M$ covering $\tau \cdot \sum m$
12:      $\mathcal{I}_{\text{keep}} \leftarrow$ TOP-K$(m, M)$
13:      **return** $\left(K[:, \mathcal{I}_{\text{keep}}, :], V[:, \mathcal{I}_{\text{keep}}, :]\right)$
14:   **else**
15:      **return** $(K, V)$                                       $\triangleright$ decode stage: keep all
16:   **end if**
17: **end function**

---

## A.3    PROMPTS FOR LV-BENCH'S DATA ENGINE

> **Role.** Act as a professional video content analyst. Describe a given video frame in English.
> **Context.** The previous frame was described as: *"{previous_description}"*. Use this as context to ensure temporal coherence.
> **Instruction.** Write a single, descriptive paragraph that:
> - Identifies the **main subject**, their specific actions, and expressions.
>
> - Describes the **environment and background**, including setting and lighting.
>
> - Highlights the **cinematic quality**, such as composition, color palette, and atmosphere (e.g., tense, serene, spectacular).
>
> **Constraints.**  Output must be **one coherent paragraph**, written in natural language prose, without bullet points or numbered lists.
> **Return.** The paragraph description of the current frame.

## A.4    LV-BENCH METRICS

### A.4.1    PRELIMINARIES: MEAN ABSOLUTE PERCENTAGE ERROR

Mean Absolute Percentage Error (MAPE) and Weighted Mean Absolute Percentage Error (WMAPE) are widely adopted evaluation metrics in forecasting (Kim & Kim, 2016), time series analysis (De Myttenaere et al., 2016), and increasingly in video quality assessment tasks (Huang et al., 2020). MAPE measures the average relative deviation between predicted values $\hat{y}_i$ and ground-truth values $y_i$, expressed as a percentage:

$$\text{MAPE} = \frac{100}{N} \sum_{i=1}^{N} \left| \frac{y_i - \hat{y}_i}{y_i} \right|. \tag{10}$$

Although simple and interpretable, MAPE can be biased when actual values $y_i$ are close to zero. To address this issue, WMAPE normalizes the absolute error by the sum of actual values, making the metric scale-invariant and more robust in practice:

$$\text{WMAPE} = \frac{\sum_{i=1}^{N} |y_i - \hat{y}_i|}{\sum_{i=1}^{N} |y_i|}. \tag{11}$$

These metrics provide interpretable percentage-based measures of consistency and prediction accuracy, and can be directly applied to quantify deviations across frames or segments in video tasks (Huang et al., 2020).

### A.4.2    VIDEO DRIFT ERROR (VDE)

Inspired by the WMAPE (Kim & Kim, 2016; De Myttenaere et al., 2016), we propose a new metric called Video Drift Error (VDE) to measure changes in video quality. The core idea involves dividing a long video into multiple smaller segments, each evaluated according to specific quality metrics (such as clarity, motion smoothness, etc). These scores are then used to calculate the relative change compared to the first segment. For long video generation, small quality deviations may accumulate within each short time segment. Over time, these deviations gradually build up (Li et al., 2025; Lu et al., 2024). This accumulation error can be quantified and detected through VDE. Specifically, a high VDE value indicates significant fluctuations or degradation in video quality as playback progresses, while a low VDE value suggests consistent quality levels throughout. Similar drift penalties have been introduced in works such as IP-FVR (Han et al., 2025), which focuses on preserving identity consistency, and MoCA (Xie et al., 2025b), which employs an identity perceptual loss to penalize frame-to-frame identity drift. Therefore, monitoring VDE during long-term video generation helps identify potential quality degradation trends and allows timely corrective actions to be taken.

Specifically, the method first divides the video into $N$ smaller segments of equal duration: $V = \{S_1, S_2, \ldots, S_N\}$, where $V$ is the full video, and $S_i$ represents the $i$-th segment.

Then the method evaluate each segment by applying a quality evaluation function (e.g., metric_function) to compute a score $Q_i$ for each segment $S_i$:

$$Q_i = \text{metric\_function}(S_i), \quad \forall i \in \{1, 2, \ldots, N\}. \tag{12}$$

Furthermore, the method compute rate of change which calculates the relative change $\Delta_i$ in quality scores from the first segment ($Q_1$) for all subsequent segments ($i \geq 2$):

$$\Delta_i = \frac{Q_i - Q_1}{Q_1}. \tag{13}$$

The final VDE value is derived as a weighted sum of absolute rate changes, using linear or logarithmic weights $w_i$:

$$\text{VDE} = \sum_{i=2}^{N} w_i \cdot |\Delta_i|. \tag{14}$$

### A.4.3 VDE METRICS

**Metric-specific VDEs.** Given the VDE shell defined in the preliminaries (reference chunk $S_1$, per-chunk scores $m_i$, and weights $w_i$), each metric instantiates $m_i$ as follows; the VDE value is then

$$\text{VDE}_{(\cdot)} = \sum_{i=2}^{N} w_i \frac{|m_i - m_1|}{m_1}, \quad w_i \in \big\{ N - i + 1, \, \log(N - i + 1) \big\}. \tag{15}$$

**VDE Clarity ($\downarrow$).** It evaluates temporal drift in image sharpness (defocus/blur). For long videos, creeping blur or inconsistent deblurring raises $\text{VDE}_{\text{clar}}$, while a low value indicates stable perceived clarity over time.

Let $f_t \in S_i$ be frames and $Y_t$ their luminance. Define per-frame sharpness by Laplacian variance and average within the chunk:

$$m_i^{\text{clar}} = \frac{1}{|S_i|} \sum_{t \in S_i} \text{Var}\big(\nabla^2 Y_t\big), \qquad \text{VDE}_{\text{clar}} = \sum_{i=2}^{N} w_i \frac{|m_i^{\text{clar}} - m_1^{\text{clar}}|}{m_1^{\text{clar}}}. \tag{16}$$

**VDE Motion ($\downarrow$).** It tracks drift in motion magnitude/smoothness (pace and jitter). Long-sequence generators often change kinetic behavior over time; a low $\text{VDE}_{\text{mot}}$ signals consistent dynamics without late-stage jitter or freezing.

Let $u_t$ denote the optical flow between consecutive frames, and define the per-frame motion energy as $E(u_t) = \|u_t\|_2$. Alternatively, one may compute a motion-smoothness score $s_t$ based on inter-frame differences. The chunk-level score is then

$$m_i^{\text{mot}} = \frac{1}{|S_i| - 1} \sum_{t \in S_i} E(u_t) \quad \text{or} \quad m_i^{\text{mot}} = \frac{1}{|S_i|} \sum_{t \in S_i} s_t, \tag{17}$$

and the final penalty is

$$\text{VDE}_{\text{mot}} = \sum_{i=2}^{N} w_i \frac{|m_i^{\text{mot}} - m_1^{\text{mot}}|}{m_1^{\text{mot}}}. \tag{18}$$

**VDE Aesthetic ($\downarrow$).** It measures drift in global visual appeal (composition, color harmony, lighting). In long videos, style can drift or collapse; low $\text{VDE}_{\text{aes}}$ indicates sustained, coherent aesthetics along the timeline.

Let $A(f_t)$ be a learned aesthetic predictor applied per frame; average within each chunk:

$$m_i^{\text{aes}} = \frac{1}{|S_i|} \sum_{t \in S_i} A(f_t), \qquad \text{VDE}_{\text{aes}} = \sum_{i=2}^{N} w_i \frac{|m_i^{\text{aes}} - m_1^{\text{aes}}|}{m_1^{\text{aes}}}. \tag{19}$$

**VDE Background ($\downarrow$).** It evaluates stability/consistency of the background (camera drift, flicker, texture boil). Long videos often accumulate spurious background motion; low $\text{VDE}_{\text{bg}}$ reflects a stable setting that does not "melt" over time.

Let $\mathbb{B}_t$ be a background mask and $u_t(x)$ the flow at pixel $x$. Define per-frame background staticness $\phi_t = \frac{1}{|\mathbb{B}_t|} \sum_{x \in \mathbb{B}_t} \mathbb{1}\big(\|u_t(x)\| \leq \tau\big)$ and average per chunk:

$$m_i^{\text{bg}} \;=\; \frac{1}{|S_i|} \sum_{t \in S_i} \phi_t, \qquad \text{VDE}_{\text{bg}} \;=\; \sum_{i=2}^{N} w_i \frac{|m_i^{\text{bg}} - m_1^{\text{bg}}|}{m_1^{\text{bg}}}. \tag{20}$$

**VDE Subject ($\downarrow$).** It captures drift in subject identity/attributes (face morphing, color/outfit changes). For long generations, identity can subtly shift; low $\text{VDE}_{\text{subj}}$ indicates the protagonist remains recognizably consistent throughout.

Let $E(\cdot)$ be a subject-identity encoder and $\bar{e}_1$ the mean embedding over subject crops in $S_1$. Define per-frame identity similarity $s_t = \cos\big(E(\text{crop}_t), \bar{e}_1\big)$ and average within the chunk:

$$m_i^{\text{subj}} \;=\; \frac{1}{|S_i|} \sum_{t \in S_i} s_t, \qquad \text{VDE}_{\text{subj}} \;=\; \sum_{i=2}^{N} w_i \frac{|m_i^{\text{subj}} - m_1^{\text{subj}}|}{m_1^{\text{subj}}}. \tag{21}$$

### A.4.4 COMPLEMENTARY METRICS

Following previous minute-long generation works (Guo et al., 2025; Cai et al., 2025), we additionally include five complementary metrics from VBench (Huang et al., 2024b) that are essential for evaluating long video generation, including: (1) *Imaging Quality*, which measures the technical fidelity of each video frame by quantifying distortions (e.g., over-exposure, noise, blur), thus reflecting the clarity and integrity of the generated imagery. (2) *Motion Smoothness*, which assesses the fluidity and realism of movements in the video, ensuring that frame-to-frame transitions are continuous and physically plausible to achieve natural motion. (3) *Aesthetic Quality*, which evaluates the visual appeal of the video frames, capturing artistic factors like composition, color harmony, photorealism, and overall beauty as perceived in each frame. (4) *Background Consistency*, which measures the stability of the scene's background across the video, determining whether the backdrop remains visually consistent throughout all frames. (5) *Subject Consistency*, which evaluates whether a subject's appearance remains consistent across every frame of the video, capturing the temporal coherence of that subject's visual identity over the entire sequence.

## A.5 OTHER NOISE SCHEDULES

For example, a simple *linear schedule* with a nonzero initial noise level is

$$\epsilon_c \;=\; \epsilon_{\min} + \frac{c}{n-1}\big(\epsilon_{\max} - \epsilon_{\min}\big), \qquad c = 1, 2, \ldots, n, \quad \epsilon_{\min} > 0, \tag{22}$$

so that the first chunk has $\epsilon_0 = \epsilon_{\min}$ (nonzero initial noise) and the last has $\epsilon_{n-1} = \epsilon_{\max}$ (maximal noise).

Similarly, a *sigmoid (logistic) schedule* grows slowly at the beginning and end, with faster change in the middle:

$$\epsilon_c \;=\; \epsilon_{\min} \;+\; \big(\epsilon_{\max} - \epsilon_{\min}\big) \frac{1}{1 + \exp\Big(-\alpha\big(\frac{c}{n-1} - 0.5\big)\Big)}, \qquad c = 1, 2, \ldots, n, \tag{23}$$

where $\alpha > 0$ controls the steepness of the curve transition (larger $\alpha \to$ sharper transition).

### A.6 LV-1.1M DATASET

To improve post-training data for semi-AR models, we introduce a private dataset named LV-1.1M, which contains fine-grained annotations for each video. We first collect videos from publicly datasets: Panda 70M (Chen et al., 2024b) and HD-VILA-100M (Xue et al., 2022), and private data (about 1M). These raw data often contain substantial amounts of noisy and low-quality material, lacking in careful curation for content quality and caption coherence. Thus, we devise several filtering criteria to select high-quality, large-motion , and long-take videos. We leverage PySceneDetect (Contributors.) to detect scene transitions and employ Q-Align (Wu et al., 2023) to remove videos with low aesthetics scores. We also use optical flow as a clue to filter out static videos with little motion dynamics. The optical flow is calculated between each pair of neighboring frames sampled at 2 fps and discard the videos with a low average optical flow score. Finally, we collect 1.1M high-quality long-take videos.

To caption them, we segment each long-take video into multiple chunks. The number of frames in each chunk is determined by the maximum input capacity of the corresponding foundational model (for example, Wan2.1 (Wan et al., 2025) allows up to 81 frames). Keyframes are extracted from each chunk and processed through GPT-4o, utilizing prompt engineering to generate captions for each individual chunk. Subsequently, GPT-4o is employed again to align all chunk-level captions, ensuring a coherent storyline throughout the entire video.

### A.7 VISUALIZATION COMPARISON

The full prompts of the Figure 2 are as follows:

"captions": [
"A serene white swan glides across a misty lake, its reflection shimmering in the calm water (00s - 03s).",
"The swan dips its head gracefully into the water,
creating gentle ripples around it (04s - 07s).",
"Lifting its head, the swan shakes off droplets, sending small splashes into the air (08s - 11s).",
"It spreads its wings slightly, flapping them to create a splash and adjust its position (12s - 15s).",
"The swan turns slightly, continuing to glide smoothly as mist hovers over the water (16s - 19s).",
"With elegant movements, the swan swims forward, its long neck curved gracefully (20s - 23s).",
"The swan pauses briefly, surveying its surroundings with a poised demeanor (24s - 27s).",
"It resumes swimming, its feathers catching the soft light filtering through the mist (28s - 31s).",
"Dipping its beak again, the swan appears to forage or drink from the tranquil waters (32s - 35s).",
"The swan lifts its head once more, shaking off water with a delicate motion (36s - 39s).",
"Turning its body, the swan reveals its full profile against the backdrop of foggy greenery (40s - 43s).",
"It continues its graceful journey, leaving a trail of ripples behind (44s - 47s).",
"The swan's reflection mirrors its every move, enhancing the peaceful ambiance (48s - 51s).",
"As it drifts further away, the swan becomes part of the misty landscape (52s - 55s).",
"The swan slows down, almost still, embodying tranquility on the quiet lake (56s - 59s)."]

As shown in Figure 2, all five baselines exhibit varying degrees of severe accumulation errors when generating minute-long videos. MAGI-1 (Teng et al., 2025), Self-Forcing (Huang et al., 2025), and PAVDM (Xie et al., 2025a) suffer from significant image quality degradation and color distortion after around 12 seconds, with the video gradually deteriorating and eventually collapsing. FramePack (Zhang & Agrawala, 2025), on the other hand, avoids severe image distortion but produces poor dynamics and limited content diversity due to its symmetric progression design. SkyReel-V2 (Chen et al., 2025) is the closest baseline in comparison, yet it still experiences noticeable color drift after 12 seconds, which continues to accumulate until the final chunk. In contrast, our method outperforms all of these approaches, maintaining subject and background consistency, preserving image quality, and preventing color degradation.

## A.8 VISUALIZATION RESULTS

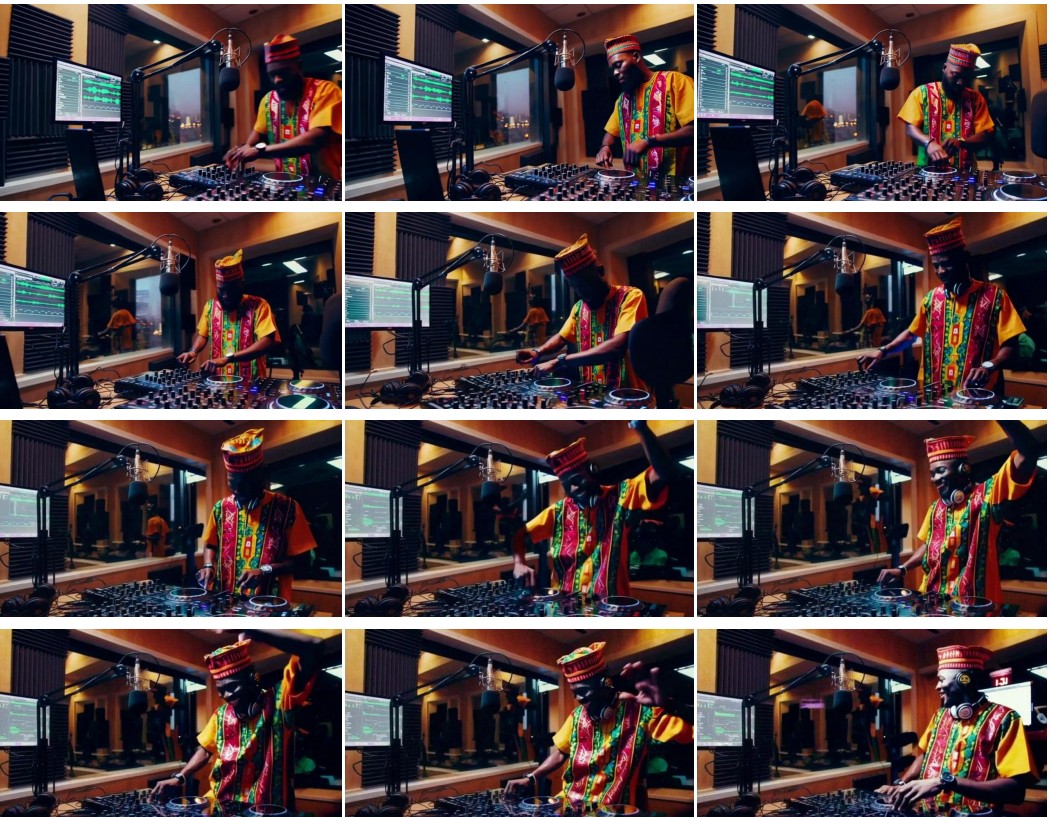

Figure 4: More visualization results #1.

"captions": [
"A DJ in vibrant Nigerian attire stands behind a mixing console, adjusting knobs with focused precision (00s - 03s).",
"He glances at the audio waveforms on his monitor, syncing his movements to the rhythm of the track (04s - 07s).",
"With smooth hand gestures, he manipulates the turntables, blending beats seamlessly in the studio (08s - 11s).",
"The DJ nods along to the music, fully immersed as he fine-tunes levels and effects (12s - 15s).",
"His reflection is visible in the glass window as he dances subtly while mixing (16s - 19s).",
"He lifts one hand in the air, hyping the unseen audience as the bass drops (20s - 23s).",
"Smiling broadly, he spins the jog wheel with flair, showcasing his technical skill (24s - 27s).",
"He raises both arms triumphantly, feeding off the energy of the music he's creating (28s - 31s).",
"Leaning into the mic, he speaks or chants rhythmically, engaging listeners through the airwaves (32s - 35s).",
"He throws his hands up again, eyes closed, lost in the groove he's crafted (36s - 39s).",
"Adjusting headphones around his neck, he continues to tweak controls with rhythmic precision (40s - 43s).",
"He gestures toward the camera with a confident smile, radiating charisma and passion (44s - 47s).",
"Moving fluidly between decks, he layers sounds with expert timing and flair (48s - 51s).",
"He laughs joyfully, clearly enjoying every moment as he commands the radio station's sound (52s - 55s).",
"Finishing his set with a final flourish, he waves to the crowd, leaving the studio buzzing with energy (56s - 59s)."]

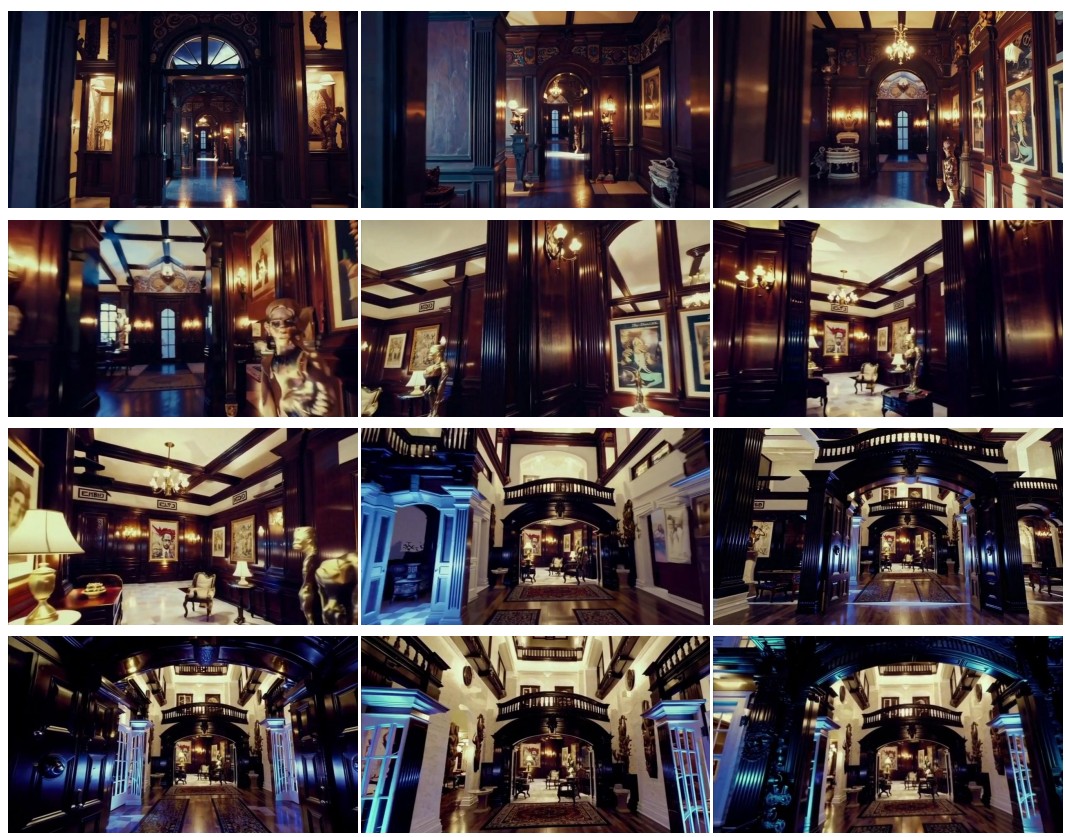

Figure 5: More visualization results #2.

"captions": [

"The camera glides through a dimly lit Victorian hallway, revealing wood paneling, arched ceilings, and warm ambient lighting (00s - 03s).",

"Classical statues and sconces line the corridor as the view advances through carved archways (04s - 07s).",

"Past gilded wall art and columns, the camera approaches a grand entryway lit by a chandelier (08s - 11s).",

"Inside a lavish sitting room, antique furniture and portraits glow under lighting (12s - 15s).",

"The camera pans across dark paneled walls with vintage posters and sculptures, highlighting the curated elegance (16s - 19s).",

"A solitary armchair beneath a chandelier, flanked by side tables and art pieces, evokes Victorian comfort (20s - 23s).",

"Rotating slowly, the camera reveals symmetrical decor — matching lamps, portraits, and ceiling beams (24s - 27s).",

"Pulling back, the view widens to a two-story foyer with a sweeping balcony and dramatic lighting (28s - 31s).",

"Double doors open to adjacent rooms, while rugs and polished floors reflect the glow (32s - 35s).",

"The camera ascends to capture the foyer's verticality with balconies, lanterns, and sculptural accents (36s - 39s).",

"Blue accent lighting outlines pillars and doors, contrasting with the warm tones (40s - 43s).",

"Through the grand archway, the viewer is drawn toward a luminous sitting area framed by columns (44s - 47s).",

"From mid-hall, the layered depth and symmetry of the mansion interior are revealed (48s - 51s).",

"Pulling back further, soaring ceilings and ornate woodwork are shown in interplay of shadow and glow (52s - 55s).",

"The final shot retreats outdoors, framing the mansion's facade at twilight, glowing windows welcoming the viewer (56s - 59s)." ]

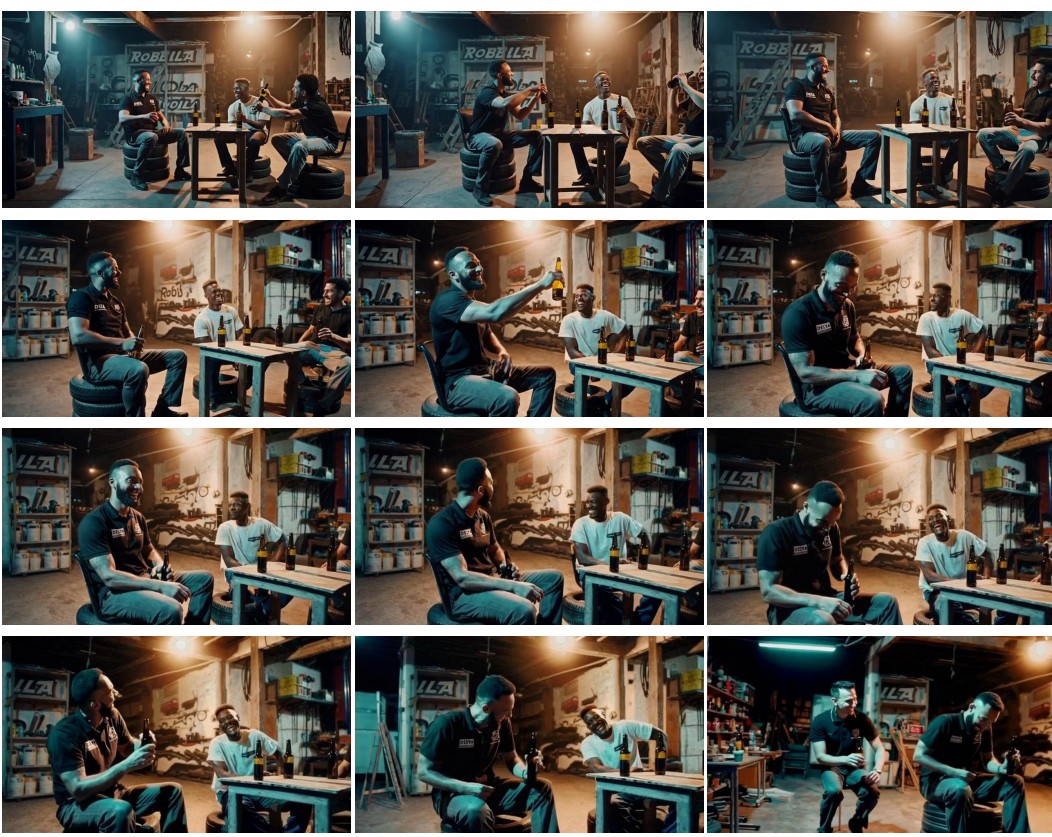

Figure 6: More visualization results #3.

"captions": [
"Three friends sit around a makeshift table in a dimly lit auto shop, clinking beer bottles in a cheerful toast (00s - 03s).",
"They laugh as they settle back into their tire seats, enjoying the camaraderie and casual atmosphere (04s - 07s).",
"The man on the left raises his bottle in a playful gesture, sharing a joke that sends everyone into laughter (08s - 11s).",
"He animatedly tells a story, gesturing with his bottle while his friends react with amusement (12s - 15s).",
"Leaning forward with a grin, he holds up his bottle triumphantly as if making a point or celebrating (16s - 19s).",
"He lowers his bottle, chuckling, clearly enjoying the moment and the company of his friends (20s - 23s).",
"Still smiling, he glances at his buddies, who are equally entertained as the vibe fills the garage (24s - 27s).",
"One friend takes a sip while the other leans back, laughing at the ongoing banter (28s - 31s).",
"The group's laughter grows louder as the man in the middle throws his head back in delight (32s - 35s).",
"The man on the left leans in again, speaking animatedly as his friends listen with attention (36s - 39s).",
"He gestures with his bottle, emphasizing his point, while the others nod and smile (40s - 43s).",
"He extends his arm to offer a bottle to his friend, sparking more laughter (44s - 47s).",
"The friends continue to enjoy each other's company, carefree amid the cluttered garage (48s - 51s).",
"A fourth friend enters, joining the laughter as the camera pans to capture the group dynamic (52s - 55s).",
"All four men share in the joyous moment, seated among tools and car parts, embodying friendship (56s - 59s)."]

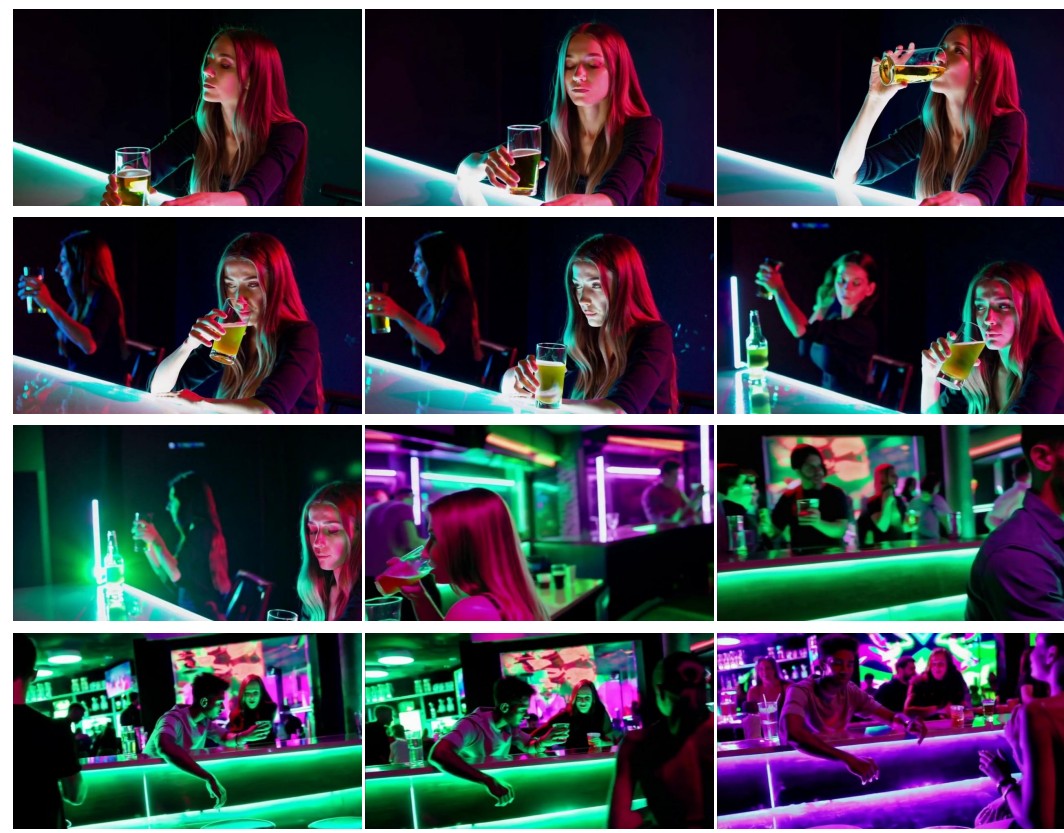

Figure 7: More visualization results #4.

"captions": [
"A young woman with long hair sits at a glowing bar, bathed in neon light, holding a glass of beer thoughtfully (00s - 03s).",
"She lifts her gaze, then brings the glass to her lips for a slow sip, the vibrant lighting highlighting her contemplative mood (04s - 07s).",
"After sipping, she lowers the glass and glances around, her reflection visible in the mirror behind the bar (08s - 11s).",
"She takes another drink while watching her reflection, neon hues shifting across her face and surroundings (12s - 15s).",
"Lowering her glass again, she looks off to the side with a pensive expression as the glow ripples over the scene (16s - 19s).",
"The camera pans slightly left, revealing glowing bottles, mirrored surfaces, and flickering colored lights (20s - 23s).",
"She takes one more sip as the background comes alive with movement — another woman dances subtly, silhouetted against the bar (24s - 27s).",
"The camera sweeps further left, showing the bustling bar environment filled with patrons under dynamic neon strips (28s - 31s).",
"A man approaches from the side, leaning in to speak as they exchange words under the pulsating lights (32s - 35s).",
"He turns away, gesturing toward the bar as other guests laugh and chat nearby (36s - 39s).",
"The camera shifts to a bartender engaging with customers while visuals flash on a screen behind him (40s - 43s).",
"He gestures animatedly, his movements synced with the rhythm of the music and lights (44s - 47s).",
"Customers smile and clink glasses as the neon-lit bar pulses with energy (48s - 51s).",
"The bartender continues his lively interaction, surrounded by the buzz of conversation and glowing lights (52s - 55s).",
"The scene ends with a wide view of the bar — people mingling, laughing, drinking — all in a neon-drenched party atmosphere (56s - 59s)."]

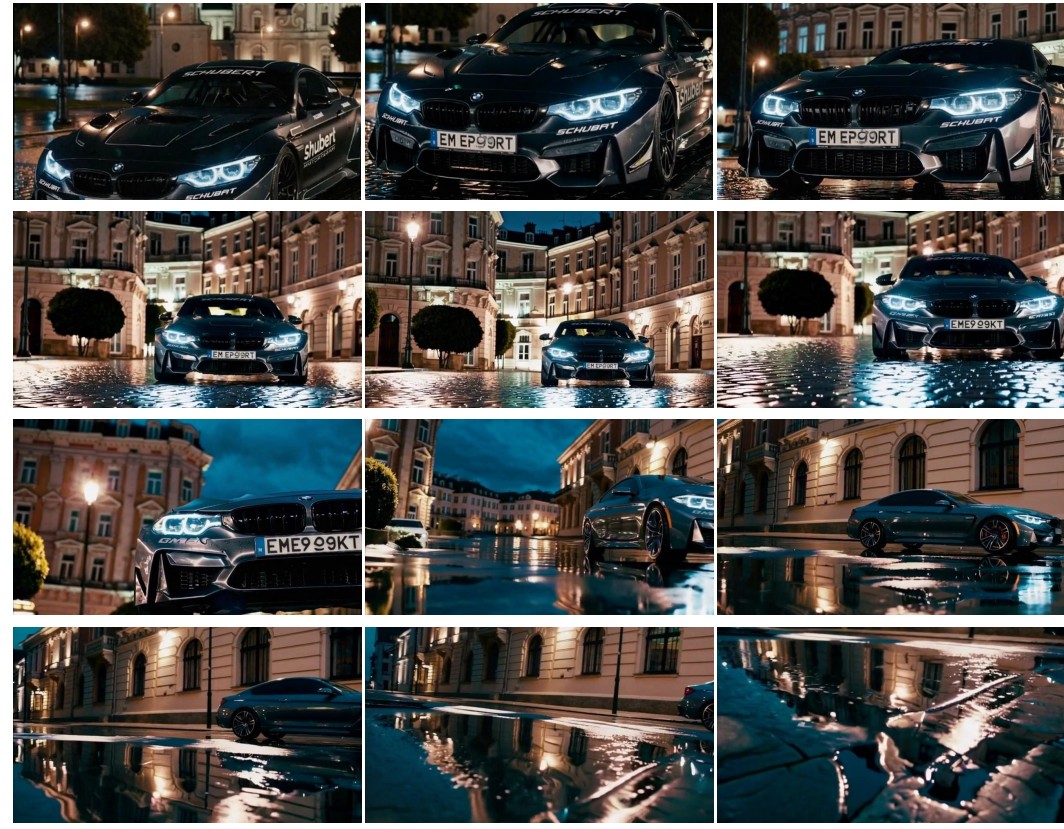

Figure 8: More visualization results #5.

"captions": [
"A sleek black BMW M4 with 'SCHUBERT' decals idles under streetlights on a wet European night, its headlights piercing the darkness (00s - 03s).",
"The camera glides closer, revealing the car's aggressive front grille and glowing LED headlights reflecting off the rain-slicked cobblestones (04s - 07s).",
"The license plate 'EM EP99RT' comes into focus as the car remains stationary, exuding power and elegance against the backdrop of historic buildings (08s - 11s).",
"The camera pulls back slightly, capturing the full front view of the BMW as it sits poised in the center of the glistening street (12s - 15s).",
"The scene widens to show the car framed by grand architecture, with ambient lighting enhancing its glossy finish and sharp lines (16s - 19s).",
"A low-angle shot emphasizes the car's stance, with reflections dancing across the wet pavement (20s - 23s).",
"The camera moves to the side, showcasing the BMW's muscular profile and intricate alloy wheels (24s - 27s).",
"As the camera sweeps along the flank, the wet street mirrors the car's silhouette and nearby street lamps (28s - 31s).",
"The shot lingers on the rear three-quarter view, capturing the interplay of light and reflection on its polished surface (32s - 35s).",
"The camera drifts lower, focusing on a shimmering puddle that reflects the car and ornate building behind it (36s - 39s).",
"The reflection becomes the focal point, blending the car's image with the glowing façade of the architecture (40s - 43s).",
"The camera glides over the reflective surface, emphasizing the serene yet powerful atmosphere of the rainy night (44s - 47s).",
"The final frames capture rippling reflections, evoking calm and sophistication as the BMW remains motionless in its urban sanctuary (48s - 52s)."]

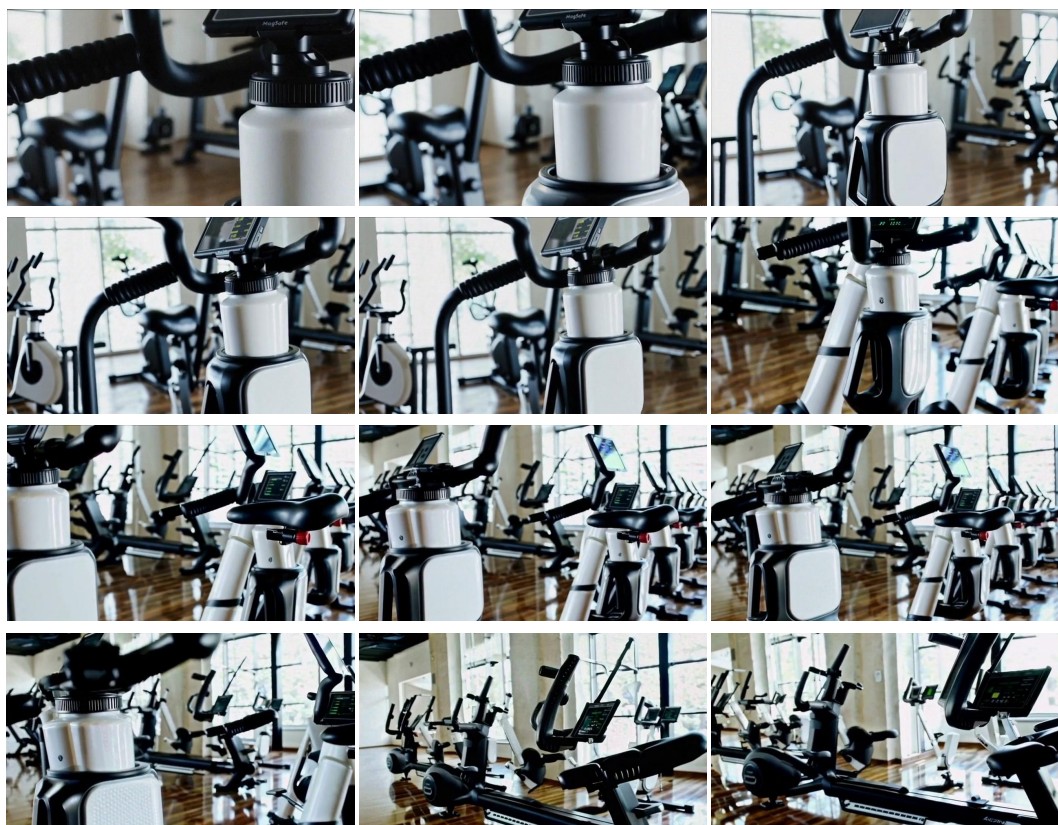

Figure 9: More visualization results #6.

"captions": [
"A close-up reveals a sleek white water bottle mounted on a black stationary bike handlebar, with blurred gym equipment in the background (00s - 03s).",
"The camera tilts down, showing the bottle and the MagSafe tablet mount (04s–07s).",
"Pulling back, the frame shows more of the bike's modern white-and-black console and handlebars against a backdrop of rows of similar bikes (08s - 11s).",
"The digital display on the bike flickers to life, showing workout metrics as the camera pans left, revealing more bikes in soft focus (12s - 15s).",
"Continuing the pan, the camera captures the rhythmic alignment of bikes and their illuminated screens, emphasizing symmetry and technology (16s - 19s).",
"The shot shifts right, focusing on the ergonomic design of the handlebars and seats, while natural light floods through large windows behind (20s - 23s).",
"Moving forward, the camera glides past multiple bikes, showcasing their clean lines and minimalist aesthetic in a spacious, polished wooden-floored gym (24s - 27s).",
"Zooming out slightly, the row of bikes extends into the distance, reinforcing the quiet, orderly, and high-tech environment (28s - 31s).", "The camera continues its slow sweep, capturing reflections on the glossy floor and the uniformity of each station's digital display (32s - 35s).",
"Pan across the gym reveals more exercise machines in the background, including recumbent bikes, all aligned under bright window-lit walls (36s - 39s).",
"Shifting focus, the camera moves toward the rear of the room, showing additional rows of black and white cardio equipment bathed in natural light (40s - 43s).",
"The perspective changes to highlight the depth of the space, with machines stretching toward distant windows, creating a sense of openness and calm (44s - 47s).",
"Gliding along the side of a recumbent bike, the camera emphasizes its sleek design and digital monitor, framed by sunlit glass panels (48s - 51s).",
"Continuing the smooth motion, the camera reveals more of the gym's layout — neat rows, reflective floors, and ambient daylight enhancing the serene atmosphere (52s - 55s).",
"Final pan showcases the full expanse of the modern fitness studio, devoid of people, radiating tranquility and technological precision (56s - 59s)." ]

## A.9 TRAINING SETUP

We first initialized our model with SkyReels-V2-DF-1.3B (Chen et al., 2025), which is a customized version of Wan2.1-T2V-1.3B (Wan et al., 2025). The video resolution used for training and inference follows the standard 480p (854×480). Experiments are conducted on a distributed computing cluster equipped with high-performance GPU nodes, each containing 192 CPU cores, 960 GB of system memory, and 8 × NVIDIA H20 GPUs (96 GB each). InfiniBand interconnects provide high-bandwidth communication across nodes for distributed training. In *Stage 1*, we train the model on 32 GPUs, requiring approximately 7 days per configuration to complete one epoch over the entire LV-1.1M dataset. In *Stage 2*, we further train the model on 32 GPUs, requiring approximately 50 hours per configuration to complete two epochs over the entire LV-Bench training set. We employ AdamW and stepwise decay schedule for all stages of post-training. The initial learning rate is $1 \times 10^{-4}$, then reduced to $5 \times 10^{-5}$, with the weight decay set to $1 \times 10^{-4}$. The noise level at the last time step corresponds to $\epsilon_{\max}$, where the SNR is 0.003, which is the default setting in Wan2.1. "In noise shuffling, we set the window size to $s = 4$, meaning that shuffling occurs among 4 frames. For the semantic sparse KV cache, due to the limitation of single-GPU memory, we use Top-$l$ semantic retrieval with $l = 2$.

## A.10 LIMITATION AND FUTURE WORK

Although our framework demonstrates strong performance in single-shot long video generation, it has not yet been thoroughly evaluated under other settings. In particular, multi-shot long video generation, where the model must handle complex transitions between multiple shots or scenes, remains unexplored. This setting introduces new challenges such as maintaining cross-shot consistency, preserving global narrative flow, and preventing semantic drift across diverse visual contexts. Addressing these challenges is critical to further extend the applicability of our foundation model.

In future work, we plan to investigate multi-shot scenarios and develop mechanisms for coherent storytelling across extended sequences. More importantly, we aim to expand LV-Bench to a larger scale and incorporate 3D-aware techniques into BlockVid to advance its capabilities as a world model. We believe expanding into these directions will further showcase the generality of our approach and contribute to building a more versatile long video foundation model.

