# OpenReview forum: "BlockVid: Block Diffusion for High-Fidelity and Coherent Minute-Long Video Generation"
_ICLR.cc/2026/Conference — ICLR 2026 Conference Withdrawn Submission_

### Official Review · Reviewer_g2VM · 2025-10-30

**Soundness:** 3
**Presentation:** 2
**Contribution:** 2
**Rating:** 4
**Confidence:** 5

**Summary:**

BlockVid aims to improve upon current block diffusion paradigm for long video generation. They propose
1. a sparse KV cache that retrieves semantically similar frame chunks to address error accumulation from Self Forcing's rolling KV cache,
2. combining Self Forcing with Velocity Forcing, which uses semantic relevant context frames as supervision,
3. progressively increasing noise schedule and noise shuffling to improve temporal consistency over the rolling long video generation, and
4. a minute-long video dataset called LV-Bench, consisting of 1000 videos and captioned every 3 seconds.

Contributions 1 and 2 are extensions to Self Forcing.

**Strengths:**

1. The LV-Bench long video dataset is a valuable contribution to the community, since such temporally captioned long video dataset is lacking.
2. Sufficient recent baselines used in comparison. BlockVid shows stronger performance than the baselines on most metrics.
3. Ablation study is comprehensive. The proposed techniques, i.e. semantic sparse KV and combining Velocity Forcing with Self Forcing, are shown to be effective. Additionally, in tab. 7 the authors show that training on minute-long videos is especially effective for generating minute-long videos, which further validates the value of the LV-Bench dataset.

**Weaknesses:**

1. Modest novelty. The proposed techniques used in the BlockVid model are mostly from previous works, i.e. Self Forcing, ZipVL, PA-VDM, and FreeNoise. Using the sementic relevant context frames is a novel contribution, which directly stems from the semantic sparse KV design. While it has been validated in the ablation study, it lacks theoretical analysis or conceptual discussions (see question 1).
2. Missing details. See questions 1, 4, 5, 7 below.
3. As acknowledge in the limitation section, this work only focuses on single-shot long video generation.

**Questions:**

1. In velocity forcing, eq. (3) replaces the GT data sample $x$ in regular flow matching loss with $\gamma \cdot x_\text{cond}$. This loss implies that, despite $x_t$ is constructed by interpolating between noise $\epsilon$ and data $x$, the model is supervised as if $x_t$ is constructed from $\epsilon$ and $x_\text{cond}$. How is $x_\text{cond}$ constructed exactly, using mathematical symbols? Why is the Velocity Forcing loss designed in this way? Why does such modification of training target work? What does the $\gamma$ coefficient control? Without supporting the design of eq. (3), it remains questionable whether such design could be applied in broader scenarios.
2. The base model SkyReels-V2 is trained with Diffusion Forcing's random noise schedule, while BlockVid is further finetuned with progressively increasing noise schedule. How do these two noise schedules compare in an ablation study?
3. Will the LV-Bench dataset and the BlockVid model be released publicly?
4. More training details. What is the context length (in terms of number of tokens, frames, chunks) of BlockVid? What are the VAE compression ratios? What is the length of the raw videos and and the clipped videos in LV-1.1M and LV-Bench? How are the per-chunk captions applied during training?
5. Most baselines, including the base model SkyReels-V2, are trained on short video data. On the contrary, BlockVid in trained on long videos from LV-1.1M and LV-Bench. This means that the long video generation evaluation requires little extrapolation from the model's training context length. It's unclear whether the proposed methods would still be as effective if BlockVid is trained on short video data like the baseline models. How does the long video data and the chunk-wise captions help compared to short, single-prompt video data? Will the proposed methods show similar improvements if BlockVid is instead trained on short video data?
6. When comparing Tab. 7 - stage 1 only with Tab. 2, the stage 1 only BlockVid model's performance is underwhelming. It underperforms many baselines, even worse than the base model SkyReels-V2. Why is this the case?
7. How are the baseline models evaluated on LV-Bench? LV-Bench video data have chunk-wise captions every 2-3 seconds. While BlockVid has been trained on such data to generate videos with interleaved prompts (Fig. 3), some the baseline models haven't been trained in this way. Are the models evaluated with chunk-wise prompts or a single concatenated prompt?
8. In addition to the improvements in quantitative results and ablation studies, how does Block Forcing and Semantic Sparse KV Cache compare to Self Forcing and Rolling KV cache? Can the authors provide some conceptual discussions and qualitative results comparing the proposed methods with Self Forcing?

I am willing to increase my rating if the authors can address my questions above.

---

### Official Review · Reviewer_Bd5Z · 2025-10-31

**Soundness:** 2
**Presentation:** 2
**Contribution:** 1
**Rating:** 2
**Confidence:** 4

**Summary:**

This paper presents BlockVid, a semi-autoregressive block diffusion framework for generating minute-long videos. It introduces a semantic sparse KV cache to reduce error accumulation, a Block Forcing strategy combining Self Forcing and Velocity Forcing to stabilize long-term generation, and progressive noise scheduling for smooth chunk transitions. The authors also propose LV-Bench, a benchmark with 1 000 annotated minute-long videos, and a Video Drift Error (VDE) metric to evaluate temporal coherence. Experiments on LV-Bench and VBench show clear improvements over state-of-the-art methods in both visual quality and temporal consistency.

**Strengths:**

1. The introduction of LV-Bench and the VDE metrics fills a real gap in assessing minute-long coherence.
2. Comprehensove ablations for noise schedule and kv cache.
3. Figures and ablations (noise schedule, KV cache, training stages) are detailed and transparent, suitable for reproduction.

**Weaknesses:**

1. The proposed components improve upon previous methods, such as sparse KV cache, self-forcing, and noise scheduling and shuffle. However, it is unclear which components are most critical for tackling the research problem of error accumulation. I recommend conducting ablation studies on each component to identify the most effective ones.
2. Considering the large training dataset of 1.1 million samples, including substantial private data, and the initialization from SkyReels-V2, direct comparisons with other methods may be unfair. I suggest conducting ablation studies on the base model and dataset to verify whether the improvements stem from the method's innovation rather than the strong baseline model and dataset.
3. Although the paper includes extensive ablation studies, it lacks analysis of the results. For example, it does not explain why the cosine noise schedule outperforms others in Table 4.
4. The supplementary visualizations are too limited, making it difficult to fully understand the performance outcomes.

**Questions:**

1. How sensitive is performance to the semantic-similarity retrieval threshold τ of retrieved caches?
2. Could BlockVid generalize to multi-scene or story-driven generation where semantic continuity is less direct?

---

### Official Review · Reviewer_gXXE · 2025-11-02

**Soundness:** 3
**Presentation:** 2
**Contribution:** 3
**Rating:** 4
**Confidence:** 5

**Summary:**

The paper proposes an autoregressive diffusion method for long video generation. Multiple components are proposed (velocity forcing, sparse kv, progressive noise, noise shuffling) to improve the self forcing baseline. It also introduces LV‑Bench and LV-1.1M datasets for training and evaluation of long video generation. The proposed method demonstrates strong performance on long video generation quality.

**Strengths:**

- The model shows impressive quality when generating long (>1min) videos, with much smaller error accumulation than prior work.
- The introduced long video dataset is a great contribution to the community.
- Ablation experiments clearly isolate gains from noise schedules, KV variants, and velocity forcing using VDE scores.

**Weaknesses:**

- The writing of the paper is overall unclear. Please see questions for things that need to be clarified.
- The motivation and description of "Velocity Forcing" is a bit unclear. In Self-Forcing style rollout, there is no ground truth chunk that the model is supposed to "reconstruct". Does Velocity Forcing take a ground-truth video as input, or does it use self-generated history? Also, the proposed loss seems to encourage the model to regenerate previously generated chunks, which does not seem to be reasonable in all scenarios, e.g., when the model is supposed to generate something entirely new. The function of the coefficient lambda is also unclear. Does lambda=1 corresponds to standard flow matching? And the model would not be learning anything when lambda=0?

**Questions:**

- Is the proposed model a few-step or a many-step diffusion model? If it's a many-step model, how does it use Self Forcing Loss?
- Which of the propose components (velocity forcing, sparse kv, progressive noise, noise shuffling) are the most important for reducing error accumulation in long video generation? From the table it seems the metrics are all better, but would be more help to show how the metrics change over the duration of generation. Also would be better to show qualitative examples in ablation studies.
- Could you describe the inference process in detail, since it seems nontrivial given KV cache and progressive noise schedule?
- What's the rough **percentage** of KV tokens that cover the fraction τ (0.97/0.98) attention score? Any evaluation on runtime/efficiency brought by the sparse attention?
- What is "Naive" in Table 6?

---

### Note · Authors · 2025-11-13

I have read and agree with the venue's withdrawal policy on behalf of myself and my co-authors.